# A Parametric Study on a Paper-Based Bi-Material Cantilever Valve

**DOI:** 10.3390/mi13091502

**Published:** 2022-09-09

**Authors:** Hojat Heidari-Bafroui, Ashutosh Kumar, Amer Charbaji, Winfield Smith, Nassim Rahmani, Constantine Anagnostopoulos, Mohammad Faghri

**Affiliations:** Microfluidics Laboratory, Department of Mechanical, Industrial and Systems Engineering, University of Rhode Island, 2 East Alumni Avenue, Kingston, RI 02881, USA

**Keywords:** paper-based sensor, bi-material cantilever, paper-based valve, fluid imbibition

## Abstract

The novel paper-based Bi-Material Cantilever (B-MaC) valve allows the autonomous loading and control of multiple fluid reagents which contributes to the accurate operation of paper-based microfluidic devices utilized for biological and chemical sensing applications. In this paper, an extensive parametric study is presented to evaluate the effects of key geometric parameters of the valve, such as paper direction, cantilever width, paper type, tape type, and sample volume, in addition to the effects of relative humidity and temperature on the functionality of the B-MaC and to provide a better understanding of the rate of fluid flow and resulting deflection of the cantilever. Machine direction, cantilever width, paper type, and tape type were found to be important parameters that affect the B-MAC’s activation time. It was also observed that the rate of fluid imbibition in the B-MaC is considerably affected by change in humidity for high (55 °C) and low (25 °C) temperatures, while humidity levels have no significant effect during imbibition in the B-MaC at an ambient temperature of 45 °C. It was also found that a minimum distance of 4 mm is required between the B-MaC and the stationary component to prevent accidental activation of the B-MaC prior to sample insertion when relative humidity is higher than 90% and temperature is lower than 35 °C. The rate of fluid imbibition that determines the wetted length of the B-MaC and the final deflection of the cantilever are critical in designing and fabricating point-of-care microfluidic paper-based devices. The B-MaC valve can be utilized in a fluidic circuit to sequentially load several reagents, in addition to the sample to the detection area.

## 1. Introduction

Microfluidic Paper-based Analytical Devices (μPADs) provide many advantages due to their usage of paper as a substrate. Paper is an inexpensive material, can be easily transported, and is biodegradable. This helps in developing inexpensive devices that use small volumes of sample and reagents. They are also easy to dispose of, are user friendly, are easy to deploy, and are compatible with biological and chemical samples. Since reagents and samples flow by capillary force through the paper substrate, a μPAD does not require pumps for fluid transport, making the paper-based device a compact self-contained system. Therefore, μPADs can be used for Point-Of-Care Testing (POCT) or On- Site Testing (OST), for immediate results. The use of these devices has been growing over the past several years due to their suitability for a variety of applications, such as infectious disease diagnosis (e.g., COVID-19 [1]), biomedical [2,3], food safety analysis [4,5,6], water analysis [7,8,9], soil analysis [10], and for many other applications [11]. Paper-based devices have also shown reproducible performances with relative standard deviation (RSD) values lower than 10% [8,12] showing good precision [13,14]. While shelf life has been considered a limitation of paper-based devices [15], on-going research is providing insights on ways to improve this [16,17].

Several different flow control mechanisms utilized in paper-based devices have been proven to allow more complex assays to be performed; most notably when numerous and sequential analytical procedures need to be accurately timed [18]. They can be carried out by actively switching the fluid flow on and off by utilizing microfluidic valve technology operating with or without an external input [19,20,21,22]. Although they generally offer users more temporal control, active flow control techniques are more difficult to build and generally necessitate the use of external instruments [23,24,25]. Passive approaches, on the other hand, often require the users to modify the system’s physical [12] and/or chemical features [26], such as the size of the transport channel [27], the viscosity of the fluid [28], or the hydrophilicity and wettability of the transport channel [29]. In our previous publications, we described a novel autonomously activated paper-based cantilever (PBC) [30] and the bi-material cantilever (B-MaC) [31,32] that offer several unique advantages for use in paper-based microfluidic devices. The B-MaC allows for the autonomous operation of paper-based devices that require sequential and timed delivery of multiple reagents. It is also simple to fabricate, requires small sample volumes to operate, and has a quick response time. The B-MaC is made up of a strip of filter paper that is partially laminated on one side with tape. Once the filter paper side is exposed to a fluid, the cellulose fibers experience hygro-expansion, whereas the adjoining tape layer does not expand and maintains its length. This difference in the expansion behavior of the paper strip and tape causes the cantilever to bend in the direction of the tape layer and come into contact with the stationary component (SC) underneath (see Appendix A). To design a μPAD employing a B-MaC as a valve mechanism to control fluid flow in the device, it is necessary to understand how the parameters associated with the geometry and ambient conditions, such as temperature and humidity, influence the fluid flow behavior and functionality of the B-MaC. In this paper, a series of experimental studies were performed to characterize the B-MaC’s performances and the fluid flow behavior in the paper according to different parameters of interest.

A variety of factors, such as solution properties (e.g., density, surface tension, and viscosity), paper material (e.g., geometric shape, porosity, permeability), and environmental conditions (e.g., temperature and humidity), generally influence the behavior of a fluid flowing in a paper-based microfluidic device [33,34]. To achieve suitable and long-lasting flow behavior for μPADs, most studies have focused on adjusting the solution characteristics and paper material [35,36,37]. Although these factors are important to specify, it is as important to also take into account how other parameters, such as temperature and relative humidity (RH), affect fluid flow in paper channels. Relative humidity (RH) is defined as the ratio of the partial pressure of vapor to the vapor pressure of water at a specific temperature [38]. In operation, μPADs are expected to experience a variety of temperature and humidity conditions, since they are designed for various applications on-site, rather than being used in a laboratory environment with consistent ambient conditions. In addition, since filter paper is naturally hygroscopic, its fluid flow characteristics will significantly vary in any humid environment. Walji and MacDonald [39] experimentally examined the effect of several factors on paper during imbibition, fed from a large fluid reservoir, such as geometry (larger width and length) and environmental conditions (humidity and temperature). However, paper-based devices are being developed for point-of-care diagnostics in which samples and reagents are generally very limited [40]. Additionally, they tested the paper strips in temperature conditions ranging from 15 to 45 °C and at a constant relative humidity of 30%. They also tested in humidity conditions ranging from 30% to 85% relative humidity but at a fixed air temperature of 20 °C. Therefore, they did not evaluate the interaction effect between variations of temperature and humidity on imbibition in paper. Castro et al. [41] described how relative humidity and channel width affect imbibition in paper-based microfluidic channels. Throughout their studies, a constant ambient temperature of 25 °C was maintained. They did not study the effect of different temperatures on liquid imbibition in paper-based channels. Furthermore, they used an adequately large reservoir of water to assume imbibition from an infinite reservoir, but many real-world uses for paper-based microfluidic devices, especially for biological applications, are constrained to limited sample volumes.

In this paper, we comprehensively investigate the effect of various material and geometrical parameters, along with the influence of surrounding conditions, on the functionality of B-MaCs. An experimental setup was designed to study B-MaC behavior as a function of cantilever width, direction of paper cut (machine or cross-machine direction), sample volume, paper type, and tape type. In addition, an environmental chamber was also designed and fabricated to study the effect of relative humidity and temperature on the rate of fluid flow of filter paper, as well as that of the B-MaC. The final deflection of cantilevers in various humidity conditions is reported on. Paper-based devices utilizing B-MaCs can benefit from this analysis of critical parameters when tests are conducted in different ambient conditions and when the sample volumes are relatively small. Besides introducing and providing two examples of the application of B-MaC as a novel valve technology in paper-based devices, proving itself to be quick to respond, easy to manufacture, and needing only small sample amounts to function, as a consequence of taking into account the interaction between temperature and humidity on the imbibition rate in filter paper, which, to the best of our knowledge, has not been fully examined in prior research investigations, the results reported in this study are beneficial to developers of paper-based devices.

## 2. Materials and Methods

### 2.1. Material

The following items were used in preparing and running the experiments presented in this paper. The paper-based valve used one of the following paper types, Whatman grade 1 chromatography paper (CHR1-GE Healthcare Whatman 1-3001878), Whatman filter papers grade 1 (GE Healthcare Whatman 1-1001824), grade 4 (GE Healthcare Whatman 4-1004917), or grade 41 (GE Healthcare Whatman 41-1441866). Tape is also an integral part of the B-MaC and one of the following tapes were used in creating the valve, Scotch^®^ Tape 600 (3M, St. Paul, MN, USA), Scotch^®^ Heavy Duty Shipping Packaging Tape (3M, St. Paul, MN, USA), and Duck Brand HD Clear High-Performance Packaging Tape (Avon, OH, USA). Table 1 and Table 2 list the properties of different paper and tape types used in this study, respectively.

ASTM Type 1 deionized water (resistivity > 18 MΩ/cm, LabChem-LC267405) was used in all experiments. A few drops of food coloring (Wilton Icing Colors) were added to this water to aid in visualizing fluid flow.

The layout of the paper-based valve was designed using vector graphics software (CorelDraw Graphics Suite X6, Corel Corporation, Ottawa, Ontario, Canada) and then cut out using a laser engraver (Epilog mini 40 W) in the paper direction of interest. An 8-megapixel video camera with 30 frames per second capability was used to capture the actuation of the B-MaC. These video recordings were then replayed using a media player (Avidemux) to collect the data.

### 2.2. Bi-Material Cantilever

A strip of filter paper (4 mm × 30 mm), partially laminated with tape (4 mm × 19 mm) on one side, leaving a 2 mm distance from the end of the paper, functioned as a cantilever actuator. While the tape layer’s length did not change, the cellulose fibers in the paper experienced hygro-expansion when exposed to fluid flow. This resulted in the free end of the cantilever bending toward the tape layer side, while the other end of the cantilever was fixed and attached to a solid support. The cantilever bent because of the difference in reaction of the paper and tape components to fluid flow, and, thus, opened or closed a fluidic channel at the stationary component (SC). The stationary component was a 1 cm × 1 cm piece of chromatography paper. The two orientations of the bi-material cantilever are shown in Figure 1. As seen in Figure 1a, when the tape layer was on the bottom side, the B-MaC bent downward as fluid started flowing from the right side. If the tape layer was attached to the top side, then the B-MaC bent upward, as shown in Figure 1b.

Fabricating a B-MaC consisted of first attaching tape to the paper. Separately, using vector graphics software (CorelDraw X6), the cantilevers’ dimensions were drawn. Then, the laser engraver was used to cut off the cantilevers from the tape and paper. The cuts were made in the cross-machine direction. The cross-machine direction was 90 degrees from the direction the pulp flowed during the paper’s fabrication.

### 2.3. Environmental Chamber

An environmental chamber was designed and built at the Microfluidic Laboratory at URI to study the effects of varying ambient conditions on the performance of the B-MaC. This chamber consisted of a test space, an electric heater, an ultrasonic atomization humidifier, humidity and temperature sensors and controllers, a cooling fan, a dehumidifying fan, a drain valve, and a vent valve. Figure 2a shows a schematic of the environmental chamber and Figure 2b shows the actual experimental setup.

The cooling fan was controlled in such a way as to push room air into the chamber to reduce the temperature inside; whereas the dehumidifying fan would suck air out of the chamber to reduce the humidity inside. The temperature and humidity in the lab were typically controlled at a comfortable indoor condition (Temp. 23 ± 1 °C and RH 50 ± 5%).

The chamber was made up of Plexi glass (3 mm thick) joined together by a few steel L-brackets and screws. A sealant (3M Marine Adhesive Sealant 4200 FC) was used along all of the inside edges to reduce air infiltration into the chamber. A mini electric heater (YOUCIDI 200 watts) was used to provide the heat required to raise the temperature inside the chamber. The humidifier used was a COOSPIDER Reptile Fogger Terrariums Humidifier to increase the relative humidity inside the chamber. The temperature sensor and controller used was the Inkbird ITC-308 Digital Temperature Controller, with a resolution of 0.1 °C and an accuracy of 1 °C. This controller was used to activate the operation of the electric heater and cooling fan in the chamber. The humidity sensor and controller used was the Inkbird Humidity Controller IHC200 with a resolution of 1% and an accuracy of 3%. This controller was used to activate the operation of the humidifier and the dehumidifying fan in the chamber. The cooling and dehumidifying fans each used a 4″ AC-RXS4. Figure 3a shows the placement of the humidity and temperature sensors inside the chamber. These sensors were placed as close to the B-MaC as possible for more representative values of humidity and temperature experienced by the valve. Figure 3b shows a fully saturated and bent B-MaC fed by a capillary tube. A borosilicate capillary glass tube with 4 mm OD and 2 mm ID was utilized to load the sample fluid onto the B-MaC sample port to provide a static pressure head.

## 3. Results and Discussion

Two main experiments were conducted in this study and are summarized in Table 3. The first one aimed to study the effect of materials and geometrical parameters on the functionality of the B-MaC, based on the one-factor-at-a-time experimental method. The influence of the paper machine direction, paper width, sample volume, paper type, and tape type were all varied and studied. The optimum parameters found in the first study were then used to fabricate the B-MaCs used in the second study. The second study involved extensive experiments run to evaluate the effect of temperature and relative humidity on the wicking behavior of the fluid and the operation of the B-MaC.

### 3.1. Effect of Key Parameters on B-MaC’s Activation Time

#### 3.1.1. Paper Direction

It has been previously demonstrated that cellulose fibers line up in a certain preferential orientation during the manufacturing process. This orientation is referred to as the Machine Direction (MD) and is the direction in which paper sheets are stretched in the direction of machine pull while they are still wet and, then, later compacted to dry [42,43]. The perpendicular direction to MD in the plane of paper is called the Cross-Machine Direction (CMD). The effect of machine direction on the wicking behavior of fluid into paper has been the subject of study by previous researchers who found contradicting results. While some studies have shown imbibition along machine direction is faster than that in cross-machine direction, since the fluid flows through the paper strip with fewer impediments due to flowing in the direction of the fibers [39,44,45], results from [41] demonstrated that even a single sheet of paper did not uniformly exhibit this favored orientation. They concluded that it is not feasible to accurately choose the MD, since it is unlikely to be the same throughout the filter paper. They also concluded that the impact of choosing MD versus CMD on the rate of fluid flow is very moderate in comparison with other uncertainties involved in paper-based device technology. Our results, however, are more in line with the former researchers, i.e., fluid flow was faster in the machine direction of paper than in its cross-machine direction, and as wicking distance increased, this disparity became more apparent.

Whatman filter paper strips, 4 mm wide and grade 41, were used in both machine and cross machine directions along with scotch tape to make the cantilevers. The time it took for the free end of the cantilever to deflect and connect to the stationary component, referred to as the activation time, was then evaluated using a 16 μL fluid sample, and the results are displayed in Figure 4 and summarized in Table 4. As shown in the figure, the cantilevers cut in the cross-machine direction demonstrated a significantly (*t*(8) = 4.445, two-tailed *p*-value = 0.002) faster activation time (*M* = 5.48, *SD* = 1.25) compared to those cut in the machine direction (*M* = 10.42, *SD* = 2.15).

Considering a cantilever as a beam, one possible explanation as to why the activation time was faster in the CMD might be attributed to the spring stiffness that depended on the geometry of the beam. Therefore, the cantilever made in cross machine direction had lower stiffness and tended to deflect more easily, since fiber orientations did not hinder its deflection.

#### 3.1.2. Cantilever Width

The impact of cantilever width on activation time was also investigated by creating cantilevers in the cross-machine direction with four different widths of 2, 3, 4, and 5 mm. We chose this range of widths since most lateral flow microfluidic devices are being developed with widths up to 5 mm [46]. Wider paper strips would also require a larger sample volume to flow in the μPAD to perform the analysis. All other parameters during this experiment were held constant, such as paper type (filter paper grade 41), tape type (Scotch), and volume of the fluid sample (16 μL). The results from this experiment are shown in Figure 5 and indicated that the width of the cantilever had a statistically significant (*p*-value < 0.01) effect on the response time of the cantilever. As the width of the cantilever increased, the end of the cantilever connected to the stationary component in a shorter period of time. Therefore, it is necessary to take into account how width affects the cantilever activation time in designing paper-based devices using B-MaCs. However, the activation time of the 4 mm wide cantilevers was not statistically significantly different than the activation time of the 5 mm wide ones. We chose the 4 mm wide cantilevers for further investigation in this study, since their activation time demonstrated a lower standard deviation than those of 5 mm.

The influence of width on the rate of fluid flow in μPADs has been previously studied and results show that wicking time decreases with increasing width, i.e., the fluid flows faster in the strip with the larger width; however, this dependence gets weaker as width gets wider [39,41,47]. It is reasonable to hypothesize that the faster response of cantilevers with wider widths is due to a higher rate of wicking in the strip.

#### 3.1.3. Sample Volume

Chen et al. [29] and Gerbers et al. [48] developed novel autonomous two- and three-dimensional microfluidic valves, based on altering the hydrophobicity of a multilayered structure by means of a surfactant. With this technique, they were able to control the order and mixing time of the sample and multiple reagents autonomously. These valves, however, were slow and required large fluid volume for actuation. The B-MaC solves this problem since it only needs a few microliters of a fluid sample to be activated. The influence of sample volume on the functionality of the cantilever was experimentally assessed by varying the sample volume from 8 μL to 20 μL in increments of 4 μL. Whatman filter paper grade 41 in cross machine direction and Scotch tape were utilized to make 4 mm B-MaCs in this experiment. The experimental results can be seen in Figure 6. A one-way analysis of variance revealed that there was not a statistically significant difference in sample volume between any of the groups (*F*(3,16) = 3.24, *F*-value = 0.75, *p*-value = 0.54). The *p*-value showed that there was no significant effect due to the sample volume. Therefore, 12 μL was selected for further investigations, since it had the lowest standard deviation.

#### 3.1.4. Paper Type

As μPADs are increasingly finding use in many different applications and more complex chemical and biological tests and assays, a proper choice of the paper type to be used is crucial. It has been experimentally shown that paper-based devices utilizing thicker papers as substrates result in slower fluid flow [49,50]. Several groups have relied on the Lucas-Washburn equation to model fluid flow in paper. This model considers paper as a bundle of capillary tubes and states that the wetted length of paper traversed by the fluid is proportional to the square root of time, surface tension, effective pore diameter, and contact angle between the liquid and the paper [51]. However, varying values for the contact angles have been previously reported [6] and in the case of the B-MaC, there are two layers of different materials with dissimilar properties. Therefore, experimentation is required to study the effect of paper on the activation time of the B-MaC. In this experiment, four commonly used papers in making μPADs [52] were employed to make cantilevers and to experimentally demonstrate the effect of paper type on the activation time of the B-MaC. Whatman chromatography grade 1, and Whatman filter paper grades 1, 4, and 41 were cut in the cross-machine direction in 4 mm widths. Scotch tape and a 12 μL sample were used to run this experiment. Figure 7 shows that the mean value of activation time was significantly different between the different paper types being tested and Whatman filter paper grade 41 presented the fastest response time. This paper type was, therefore, selected for the remaining tests.

#### 3.1.5. Tape Type

The commonly available Scotch^®^ Tape 600 (3M, St. Paul, MN, USA), Scotch^®^ Heavy Duty Shipping Packaging Tape (3M, St. Paul, MN, USA), and Duck Brand HD Clear High-Performance Packaging Tape (Avon, OH, USA) were utilized to produce cantilevers and examine the effect of different tapes on the B-MaC activation time. The cantilevers were made from Whatman filter paper grade 41 in cross-machine direction and then cut in 4 mm widths. The tests were run using 12 μL of sample. As indicated in Figure 8, although there was not a statistically significant difference between the performance of cantilevers made from heavy duty Scotch tape and Duck tape, cantilevers made from regular Scotch tape showed significantly different results.

Table 5 summarizes the design aspects of the paper-based cantilever to show the results of the influence of geometry and material parameters on the activation time of B-MaCs.

### 3.2. Effect of Temperature and Humidity

The effect of temperature and humidity on the wetted length of filter paper and B-MaC was extensively investigated. Graphical results are provided to study the relationship between the dependent variable and independent variables. The wetted length of the filter paper on the B-MaC was set to be the dependent variable and temperature and humidity were set to be the independent variables. Results for the study were obtained for four temperature settings, 25 °C, 35 °C, 45 °C, and 55 °C, with a corresponding relative humidity value ranging from 10% to 90%, with an incremental step of 20%.

#### 3.2.1. Imbibition through Filter Paper Strips

Figure 9 shows the relationship between the wetted length of filter paper, over a period of time, for different temperatures and humidity levels. The results show that the imbibition of fluid though filter paper remained constant at low temperatures irrespective of the humidity level. However, this trend changed with an increase in temperature.

Figure 9a demonstrates that at the low temperature setting of 25 °C the wetted length for filter paper remained unaffected for humidity settings of 10% RH, 30% RH, 50% RH, 70% RH, and 90% RH. However, raising the temperature from 25 °C to 35 °C, resulted in the wetted length of the filter paper changing significantly with changes in humidity. For instance, the rate of change of the wetted length was slowest for low humidity and highest for high humidity. Figure 9b shows the rate of change of wetted length for humidity settings of 10% RH, 30% RH, 50% RH, 70% RH, and 90% RH, at a temperature setting of 35 °C. The rate of change of wetted length gradually increased from low (10% RH) to high humidity (90% RH) level. Increasing the temperature from 35 °C to 45 °C resulted in further decreasing of the wetted length as a function of time. It also increased the variability. Figure 9c shows that at 45 °C the average time to achieve a 40 mm wetted length was reduced for all humidity levels. Furthermore, at 30% RH and 50% RH, the wetted length for the filter paper was unchanged, indicating an intermediate trend transformation; however, it was hard to justify the trend due to the limitation of the experiment. During the experiment, the setting of 55 °C and 90% RH was difficult to achieve and stabilize which led us to remove it, due to the uncertainty in trend analysis for the wetted length of filter paper. Figure 9d demonstrates that the average time for fluid imbibition in filter paper was further reduced with an increase in temperature from 45 °C to 55 °C.

In summary, the fluid imbibition in filter paper at low temperature was not affected by the humidity level, but for higher temperatures, the time to reach a wetted length of 40 mm decreased. To the best of our knowledge, the effect of relative humidity and temperature on fluid flow behavior in paper strips has so far only been characterized by two research groups. Nevertheless, their results appeared to contradict each other on the effect of humidity on the imbibition flow in filter paper strips. While Walji and MacDonald [39] observed that the effect of humidity at room temperature on the imbibition speed was negligible, the report from Castro et al. [41] showed that the imbibition speed increases with higher humidity levels at a fixed temperature of 25 °C. Both groups, however, did not consider the interaction effect between relative humidity and temperature. Although our results in Figure 9a proved the results from [39] and were not in line with [41], Figure 9b–d show that by increasing the surrounding temperature, the relative humidity did, in fact, matter in the length of wetted portion of paper strips, because as relative humidity rose, the imbibition distance increased at a given time. For further assessment, a comparison is provided in Table 6 between the effect of paper direction, paper width, temperature, humidity, and interaction between temperature and humidity on imbibition in paper-based microfluidic channels presented in this work and those obtained by previously reported studies.

#### 3.2.2. Estimation of the Wetted Length of B-MaC

Since the cantilever bends with the flow of fluid, it is more difficult to experimentally measure the wetted length of cantilevers over time. For this purpose, the B-MaC was set against the graduated wall with a resolution of 1 mm by 1 mm. The experiment was conducted by placing the B-MaC on a fixed component and loading fluid onto the B-MaC with a capillary tube, as was presented in Figure 3b. As the fluid was being loaded onto the B-MaC, the moving fluid front, and the wetted length, were captured with the aid of a video camera. The values recorded for time taken to achieve the desired wetted length of B-MaC ranged from 0 mm to 40 mm with a step of 5 mm, until the B-MaC reached a stationary position. This stationary position is the maximum deflection that can be obtained from any given B-MaC on fluidic loading. The x and y coordinates of the end of wetted position were also recorded during bending the B-MaC and then plotted in a “y vs. x” graph. Figure 10 displays an example of a bending cantilever at a specific surrounding condition, namely, at T = 25 °C and 10% humidity. Exponential and second-order polynomial curves were fitted to the data, and it turned out that the second-order polynomial curve fit well with an R^2^ = 0.9988. Considering this fact, the wetted lengths of B-MaCs were calculated by a line integral using the following formula:(1)$$L=\int_{a}^{b}ds\;\;\;\;where\;\;\;ds=\sqrt{{\left(\frac{dx}{dt}\right)}^{2}+{\left(\frac{dy}{dt}\right)}^{2}}dt$$

#### 3.2.3. Imbibition through B-MaC

Figure 11 depicts the relation between the length of B-MaC that has been wetted over time at various temperatures and levels of humidity. Figure 11c demonstrates that at a temperature setting of 45 °C, the wetted length of a filter paper remained unaffected for humidity settings of 10% RH, 30% RH, 50% RH, 70% RH, and 90% RH. With an increase in temperature from 45 °C to 55 °C the wetted length on filter paper changed significantly with the range of humidity. For instance, rate of change of wetted length was slowest for high humidity and rate of change of wetted length was highest for low humidity for B-MaC. As stated above, due to the limitation of the experimental setup, the setting of 55 °C and 90% RH was difficult to achieve and stabilize so it was not included in the analysis. Figure 11d demonstrates that the average time for fluid imbibition in B-MaC reduced on increase in temperature from 45 °C to 55 °C for high humidity and the average time for fluid imbibition in B-MaC increased on increase in temperature from 45 °C to 55 °C for low humidity. On the contrary, Figure 11b demonstrated that the average time for fluid imbibition in B-MaC reduced on decrease in temperature from 45 °C to 35 °C for low humidity and the average time for fluid imbibition in B-MaC increased on decrease in temperature from 45 °C to 35 °C for high humidity. In comparison, the fluid imbibition though B-MaC at 55 °C, compared to that at 25 °C, clearly showed that the trend was reversed for wetted length versus time going from a low temperature of 25 °C to a high temperature of 55 °C. Figure 11a demonstrates that the average time for fluid imbibition in B-MaC at 25 °C was highest for high humidity and lowest for low humidity.

In summary, the fluid imbibition in B-MaC at 45 °C temperature was not affected by the humidity level. However, on an increase in temperature, the average time to achieve a wetted length of 25 mm increased with a decrease in humidity, and on a decrease in temperature, the average time to achieve a wetted length of 25 mm increased with an increase in humidity. It can be concluded from the graphical results that the wetted length for B-MaC was drastically affected by a change in humidity for high temperatures.

### 3.3. Effect of Temperature and Humidity of B-MaC Deflection

Activation of the B-MaC is based on the difference between the elongation of paper and that of tape when paper gets wetted. However, the paper part of the B-MaC can also absorb moisture from the ambient surroundings and cause the B-MaC to start bending even without introducing any sample to it. Thus, the effect of temperature and relative humidity should be carefully studied to know the behavior of B-MaCs as a component in microfluidic paper-based devices. We recorded the initial deflection of B-MaC while in the equilibrium state within the environmental chamber after 10 min and before the introduction of the fluid sample. The B-MaC’s final deflections were also recorded when fully wetted by the fluid sample. The difference between the B-MaC’s final and initial deflections was referred to as the net deflection. Figure 12 shows that a B-MaC does bend under different environmental conditions. As the left inset in Figure 12 indicates, the B-MaC could be stable and straight in lower humidity, but as humidity increased, the B-MaC started bending after absorbing moisture from the environment. However, by raising the temperature, the initial deflection of the B-MaC was reduced due to evaporation of the moisture content in the paper. On the other hand, the final deflection, see the right inset in Figure 12, had a lower dependency on relative humidity and varied between 6.5 to 8.5 mm after the B-MaC was fully wetted by the fluid sample. The data in Figure 12 indicates that a minimum distance of 4 mm was required between the B-MaC and the stationary component to prevent inadvertent activation of the B-MaC before sample application for the different temperatures and humidity levels under consideration, after the B-MaC was equilibrated with its environment.

### 3.4. Application of the B-MaC in Microfluidics Devices

The B-MaC offers an inexpensive and simple-to-fabricate actuator system that can be easily integrated in 2D or 3D microfluidics paper-based devices. The bi-material actuator may also be used in complicated systems, like the enzyme-linked immunosorbent assay (ELISA) protocols, where many reagent fluids are involved, and sequential loading or timing delays are necessary. Figure 13, for instance, demonstrates the use of a partitioned B-MaC to stop fluid flow through a stationary component after a given time has elapsed. That time is determined by the duration of the timing-fluid flow through the π structure shown in the figure and actuation of the B-MaC. The white arrows in the figure indicate the fluid flow direction.

Figure 14 shows a three-dimensional paper-based device with two B-MaCs that allow the sequential loading of two reagents onto the test zone. Two reagents were previously loaded into their respective pads prior to sample addition (Figure 14a). The sample then flowed in both directions and B-MaC 1 was activated first (Figure 14b). Once B-MaC 1 was engaged with SC 1, the reagent 1 (green) flowed through the test area as well as in the delay channel under B-Mac 2 (see Figure 14c). As seen in Figure 14c, the green reagent was visible in the last leg of the delay channel, as it activated B-Mac 2 at about 3 min. The engagement of B-MaC 2 caused reagent 2 (red) to start flowing toward the test zone. Reagent 2 arrived at the test area at about 8 min. The test was allowed to run, and a strong signal developed, as shown in Figure 14d.

## 4. Conclusions

In this work, the effect of various material and geometrical parameters used in the B-MaC, such as direction of paper to be cut, paper width, sample volume, tape type, and paper type, on its activation time were studied experimentally. Test results showed that all of these parameters, aside from sample volume, play a significant role in the B-MaC’s activation time and an optimized value for each parameter was provided to lower its activation time. These optimized values were then used in the design of an experiment to study the combined effect of relative humidity and temperature on the rate of fluid flow in filter paper, as well as their effect on the B-MaC in an environmental chamber specifically built for that purpose.

Results showed that the fluid imbibition in filter paper at a temperature of 25 °C was not affected by the different humidity levels examined in the study. This was not the case for higher temperatures where humidity had an effect on fluid flow and the average time to achieve a wetted length of 40 mm decreased with increasing temperature. The level of humidity had an effect on the imbibition rate in the B-MaC at different temperatures and this could be attributed to the difference in response of the paper and tape layers in the B-Mac. The study also found the net deflection of the B-MaC in various environmental conditions. Finally, a recommendation was made on the minimum distance required between the cantilever and the stationary component to prevent the inadvertent activation of the B-MaC in various environmental conditions.

The results presented in this paper are of value to designers of paper-based devices, since the results take into account the interaction of temperature and humidity on the imbibition rate in filter paper which, to the best of our knowledge, has not been exhaustively looked at in previous research studies. Designers of paper-based devices utilizing B-MaCs in their architecture will also benefit from this study as it provides optimized values for the different geometric parameters of interest when designing the B-MaC. Several reagents and the sample can be loaded successively into the detecting region using the B-MaC valve in a fluidic circuit.

## Figures and Tables

**Figure 1 micromachines-13-01502-f001:**
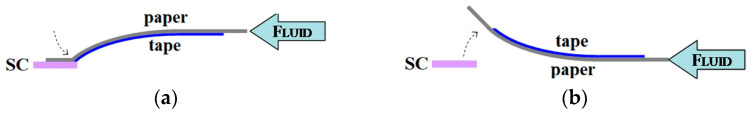
Schematic diagram of bi-material cantilever operating in two configurations. (**a**) Fluid flow connects the cantilever to the SC. (**b**) Fluid flow disconnects the cantilever from the SC.

**Figure 2 micromachines-13-01502-f002:**
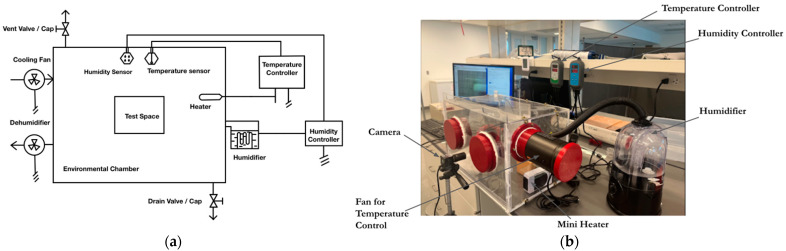
(**a**) A schematic representation of the environmental chamber showing the different components used. (**b**) Actual experimental set up used in this study.

**Figure 3 micromachines-13-01502-f003:**
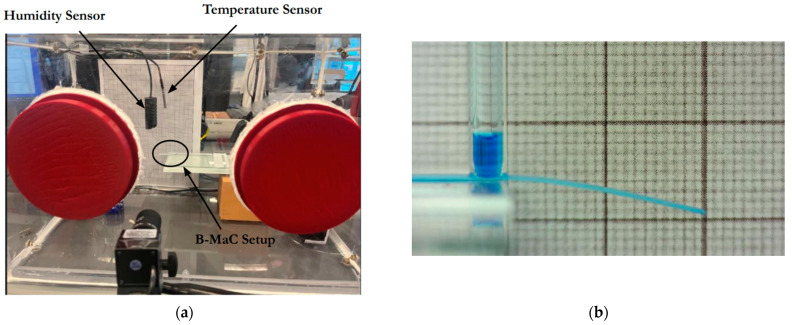
(**a**) Position of mounted humidity and temperature sensors. (**b**) Usage of a capillary tube to provide a consistent sample volume for each test run.

**Figure 4 micromachines-13-01502-f004:**
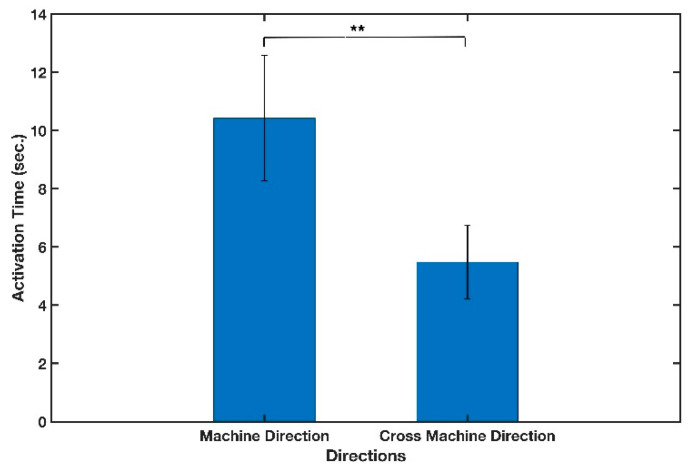
Effect of paper direction on activation time of cantilever. Mean values were obtained from five different samples (n = 5) per condition. The error bars represent the standard deviation. Two-sample *t*-test was performed on the data (** *p*-value < 0.01).

**Figure 5 micromachines-13-01502-f005:**
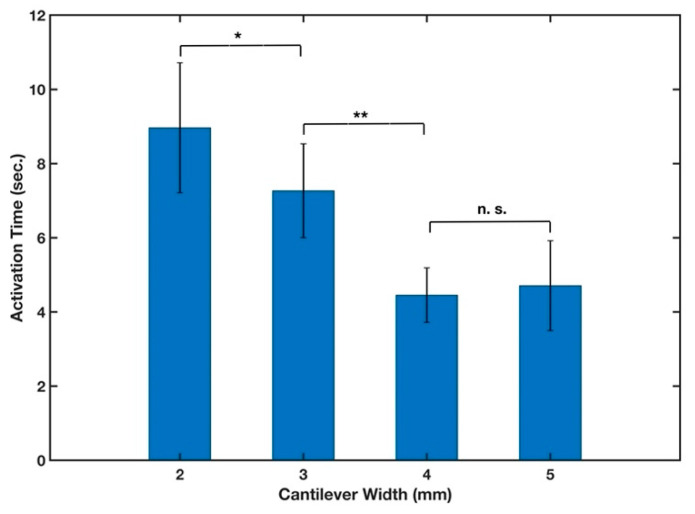
Effect of cantilever width on the activation time of the cantilever. n = 5 and the error bars represent the standard deviation. Two-sample *t*-tests were performed on the data (* *p*-value < 0.05, ** *p*-value < 0.01, and n. s. stands for not significant).

**Figure 6 micromachines-13-01502-f006:**
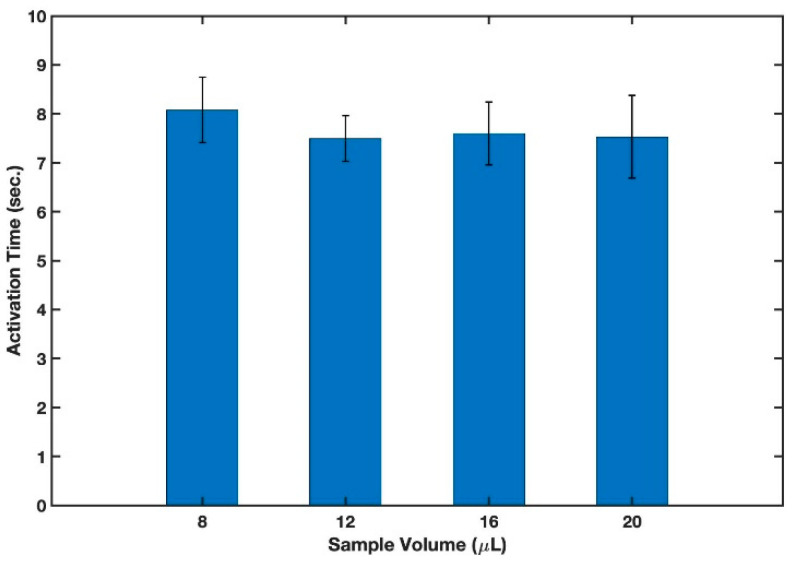
Effect of sample volume on activation time of cantilever. n = 5 and the error bars represent the standard deviation.

**Figure 7 micromachines-13-01502-f007:**
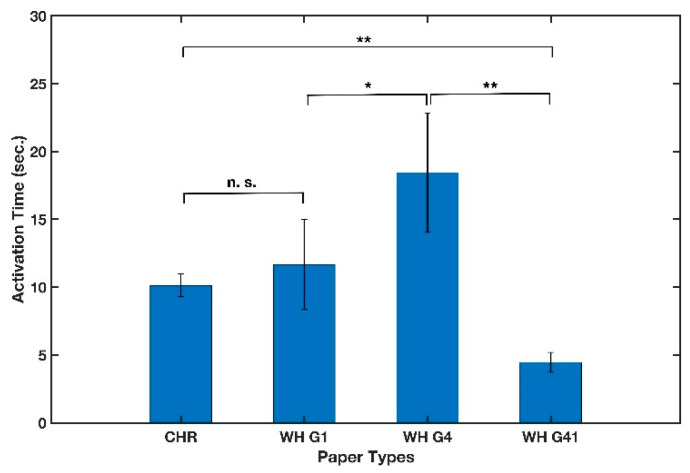
Effect of paper type on activation time of cantilever. n = 5 and the error bars represent the standard deviation. Two-sample *t*-tests were performed on the data (* *p*-value < 0.05, ** *p*-value < 0.01, and n. s. stands for not significant).

**Figure 8 micromachines-13-01502-f008:**
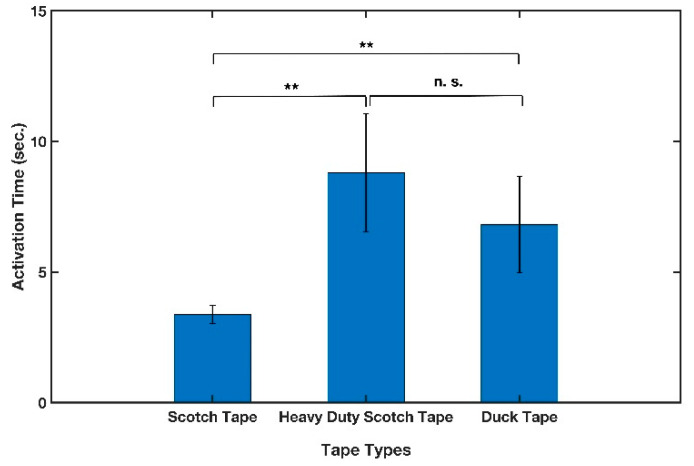
Effect of tape type on activation time of cantilever. n = 5 and the error bars represent the standard deviation. Two-sample *t*-tests were performed on the data (** *p*-value < 0.01, and n. s. stands for not significant).

**Figure 9 micromachines-13-01502-f009:**
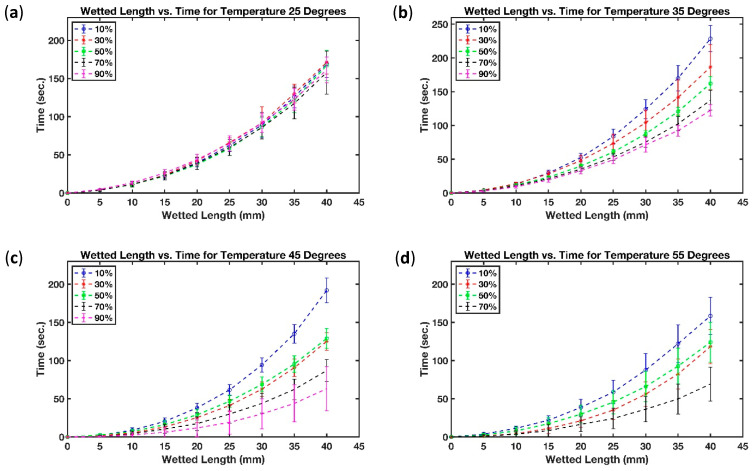
Effect of relative humidity on the wetted length of paper strips at selected temperatures (**a**) T = 25 °C (**b**) T = 35 °C (**c**) T = 45 °C and (**d**) T = 55 °C.

**Figure 10 micromachines-13-01502-f010:**
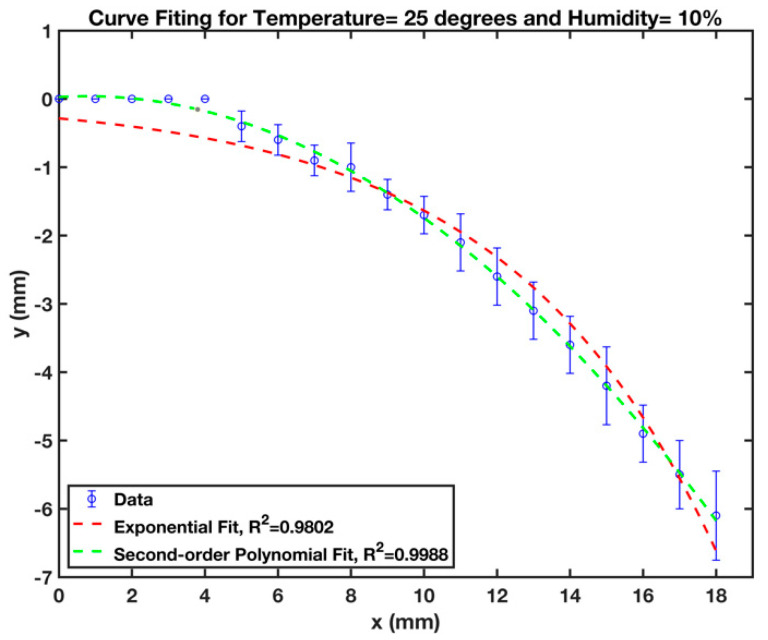
Curve fitting on bended B-MaCs at T = 25 °C and 10% humidity.

**Figure 11 micromachines-13-01502-f011:**
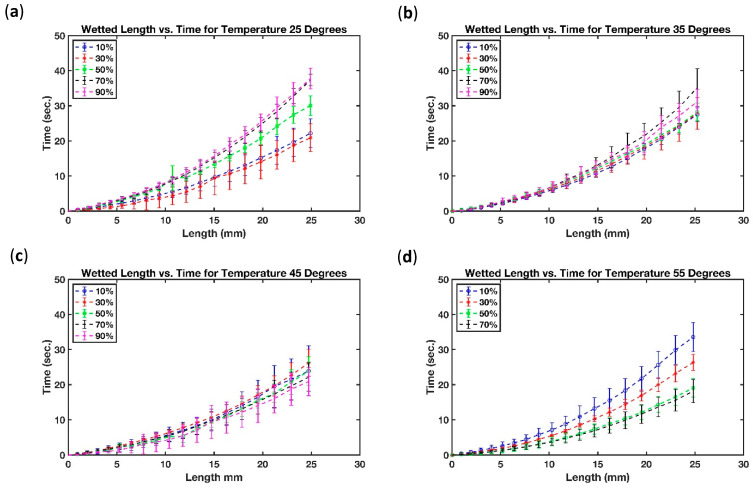
Effect of relative humidity on the wetted length of B-MaCs at selected temperatures (**a**) T = 25 °C (**b**) T = 35 °C (**c**) T = 45 °C and (**d**) T = 55 °C.

**Figure 12 micromachines-13-01502-f012:**
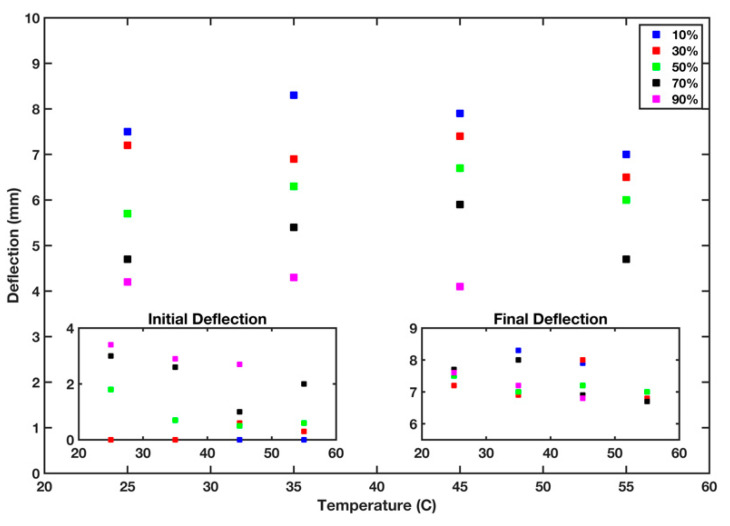
Initial, final, and net deflection of B-MaC at different temperatures and relative humidity conditions.

**Figure 13 micromachines-13-01502-f013:**
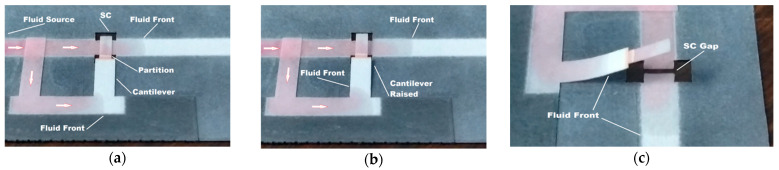
Demonstration of the B-MaC in a time delay circuit: (**a**) The B-MaC engaged with SC and allowing flow, (**b**) Timing fluid raised the B-MaC and stopped flow through the SC, (**c**) Improved view of the fluidic disconnect.

**Figure 14 micromachines-13-01502-f014:**
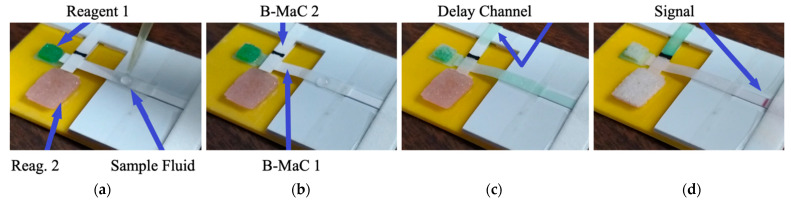
A three-dimensional paper-based device with two B-MaCs that allow the sequential loading of two reagents after sample addition. (**a**). Adding sample fluid, (**b**). Activation of the first B-MaC by the sample fluid, (**c**). Activation of the second B-MaC by the first reagent fluid, (**d**). The signal in the detection zone.

**Table 1 micromachines-13-01502-t001:** Properties of different paper types used in this study.

Paper Type	Thickness (μm)	Average Pore Size (μm)	Basis Weight (g/m^2^)	Parameter
Whatman grade 1 chromatography	180	11	87	130 mm/30 min flow rate
Whatman filter papers grade 1	180	11	87	0.25 psi wet burst 150 s/100 mL speed (Herzberg)
Whatman filter papers grade 4	210	20–25	92	0.22 psi wet burst 37 s/100 mL speed (Herzberg)
Whatman filter papers grade 41	220	20–25	85	0.22 psi wet burst 54 s/100 mL speed (Herzberg)

**Table 2 micromachines-13-01502-t002:** Specifications of three different tapes utilized in this study.

Tape Type	Thickness (μm)	Tensile Strength (N/cm)	Adhesive
Scotch^®^ Tape 600	58	49	Pressure sensitive acrylic
Scotch^®^ Heavy Duty Shipping Packaging	78	82	Thermosetable Rubber Resin
Duck Brand HD Clear High-Performance Packaging Tape	66	54	Pressure sensitive acrylic

**Table 3 micromachines-13-01502-t003:** A summary of the two main categories of experiments conducted in this work.

Experiment	Fixed Conditions	Variable Conditions	Method
Experiment 1	Temperature and Humidity	Paper machine direction, paper width, sample volume, paper type, and tape type	one-factor-at-a-time
Experiment 2	Paper machine direction, paper width, sample volume, paper type, and tape type	Temperature and Humidity	two-factor factorial design

**Table 4 micromachines-13-01502-t004:** A summary of conducted test variables and results.

Independent Variable (Paper Direction)	Dependent Variable (Activation Time)	Fixed Variables	T-Test Result
Machine Direction	*Mean* = 10.42 s. *SD* = 2.15 s.	Cantilever Width = 4 mm Sample Volume = 16 μL Paper Type = Whatman Grade 41 Tape Type = Scotch Tape	*t*(8) = 4.445, *p*-value = 0.002
Cross-machine Direction	*Mean* = 5.48 s. *SD* = 1.25 s.		

**Table 5 micromachines-13-01502-t005:** Summary of the geometric parametric study on activation time of B-MaCs.

Parameter	Levels	Values	Results
Paper direction	2	Machine and Cross Machine	Significant difference- cross machine direction had the lowest activation time.
Cantilever width	4	2, 3, 4, and 5 (mm)	Significant difference- 4 and 5 mm had the lowest activation time- 4 mm was selected.
Sample volume	4	8, 12, 16, and 20 (μL)	No difference- 12 μL was selected.
Paper type	4	CHR 1, WH G1, WH G4, and WH G41	Significant difference- WH G41 had the lowest activation time- WH G41 was selected.
Tape type	3	Scotch, HD Scotch, and Duck HD	Significant difference- Scotch tape had the lowest activation time- Scotch tape was selected.

**Table 6 micromachines-13-01502-t006:** Comparison of results on the effect of different parameters on fluid flow in paper-based devices with previously reported studies.

Ref.	Paper Direction	Paper Width	Temperature	Humidity	Interaction between Temperature and Humidity
[39]	Significant	Significant	Significant	Not Significant	N/A
[41]	Not significant	Significant	N/A	Significant	N/A
This work	Significant	Significant	Significant	Not Significant	Significant

N/A: Not Applicable.

## Data Availability

Data is contained within the article. Additional data not presented in this article is available on request from the corresponding author.

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
