# Peer review of "A Parametric Study on a Paper-Based Bi-Material Cantilever Valve"

_micromachines, 2022, doi:10.3390/mi13091502_

Round 1
Reviewer 1 Report
This paper reports about the paper-based cantilever which can be utilized for biological and chemical sensing applications. Here, an extensive parametric study is presented to see the effects of geometric parameters. However, the manuscript is not well-prepared and requires a complete revision.
Detailed comments are as follows:
1. What are the reliability aspects of paper-based devices? Please incorporate in the manuscript.
2. Please highlight the novelty aspects of the present research work in the manuscript.
3. Provide a comparative table/chart and compare these results with state-of-art microfluidic paper-based analytical devices.
4. Figure 4: What is the meaning of (*p < 0.05, **p < 0.01), not clear?
5. Figure14. figure caption should be elaborated.
6. How does the wetted length of cantilevers was measured/estimated?
7. Consider the design aspect of paper-based cantilevers and incorporate in the manuscript.
Reviewer 2 Report
My comments are attached as pdf.

Round 2
Reviewer 1 Report
Comments are answered satisfactorily and may be published.